# The Performance Evolution of Match Play Styles in the Spanish Professional Basketball League

**Miguel-Ángel Gómez** [1,2] , **Ramón Medina** [3] , **Anthony S. Leicht** [2] , **Shaoliang Zhang** [4,*] **and Alejandro Vaquera** [3,5]

1   Facultad de Ciencias de la Actividad Física y del Deporte, Universidad Politécnica de Madrid, 28040 Madrid, Spain; miguelangel.gomez.ruano@upm.es
2   Sport and Exercise Science, James Cook University, Townsville 4814, Australia; anthony.leicht@jcu.edu.au
3   Faculty of Physical Activity and Sport Sciences, University of Leon, 24007 Leon, Spain; rmedig00@gmail.com (R.M.); a.vaquera@uleon.es (A.V.)
4   Division of Sport Science & Physical Education, Tsinghua University, Beijing 100084, China
5   Institute of Sport and Exercise Science, University of Worcester, Worcester WR2 6AJ, UK
*   Correspondence: zslinef@mail.tsinghua.edu.cn

**Abstract:** The aim of this study is to analyse the performance evolution of all, and the dominant, team's performances throughout an eight-season period within the Spanish professional basketball league. Match-related statistics were gathered from all regular season matches (n = 2426) played during the period 2009–2010 to 2016–2017. The non-metric multidimensional scaling model was used to examine the team's profiles across seasons and for the most successful (playoff) teams. The main results showed that: 3-point field goals made (effect size, d = 0.61; 90% confidence interval, CI = 0.23; 1.37) and missed (d = 0.72; 90% CI = 0.35; 1.46), and assists (d = 1.27; 90% CI = 0.82; 1.86) presented a positive trend with an increased number of actions across the seasons; 2-point field goals made (d = 0.21; 90% CI = −1.25; 2.02) and missed (d = 0.27; 90% CI = −0.52; 0.92) were decreased; free throws made and missed, rebounds, fouls, blocks, steals and turnovers showed a relatively stable performance. The matrix solution (stress = 0.22, rmse (root mean squared error) = $7.9 \times 10^4$, maximum residual = $5.8 \times 10^3$) indicated minimal season-to-season evolution with the ordination plot and convex hulls overlapping. The two most dominant teams exhibited unique match patterns with the most successful team's pattern, a potential benchmark for others who exhibited more dynamic evolutions (and less success). The current findings identified the different performances of teams within the Spanish professional basketball league over eight seasons with further statistical modelling of match play performances useful to identify temporal trends and support coaches with training and competition preparations.

**Keywords:** team sports; performance indicators; long-term trends; performance patterns; match-related statistics; non-metric multidimensional scaling

## 1. Introduction

Statistical modelling in team sports has been an important analytical tool to predict match outcome and the evolution of team's performance [1–4]. In particular, examination of match-related performance indicators in basketball has enabled modelling to identify winning and losing teams [5], best and worst teams [6], the influence of contextual-related variables [7], the effect of consecutive games [8] or the impact of rule changes [9–11] on team's performances. Typically, these analyses have been conducted over a season [5,7], however, analysing team performances over many seasons may identify the evolution and impact of changes on a team's style of play [11]. Specifically, the analysis of an entire



competition and its most successful teams may reveal the key indicators to winning, including team adaptations for success [6]. Several studies have identified 2-point field goals made and/or defensive rebounds and/or assists as the most prominent team indicators that result in winning basketball matches [5,12–14]. However, the style of basketball match play has likely evolved due to changes in rules (i.e., an increase in the 3-point line distance), match pace (i.e., greater number of ball possessions per match), and team focus on particular match indicators such as turnovers, 3-point field goals attempted or fouls [10,11,15]. A greater understanding of how basketball performances have evolved, particularly those teams that have adapted behaviours to win consistently over many seasons [16], would assist teams for greater and long-term success.

To our knowledge, only one study has examined the evolution of basketball performances, albeit during one season [17]. This study examined all National Basketball Association teams during the 2016–2017 season and via non-metric multidimensional scaling (nMDS) identified that: (i) team playing profiles were similar at the beginning and end of the season; (ii) each team displayed a unique evolution of performance throughout the season; (iii) there was an increased number of 3-point field goals made by all teams as the season progressed. This unique analysis was limited to one season only with examination of longer-term trends and team's performances needed to elucidate the evolution of basketball match play and the importance of team's playing styles [16,17].

Prior studies have employed multivariate modelling such as nMDS to investigate match play differences across seasons and/or competitions in various sports such as Netball [18], Rugby League [19] and Australian Football [16]. This non-linear approach has clearly shown the idiosyncrasy of teams' performances and provided a unique examination and visualization of sporting evolution [16,18,19]. Further, it has identified team adaptations employed by successful teams that could be utilised by others to enhance future success [18,19]. However, no such long-term examination has been conducted in professional basketball. Clarification of the profiling of team performances over an extended period would support coaches and organisations to develop future playing styles for success. Therefore, the aims of this study were to: 1) analyse the evolution of elite basketball team's performances over time within the Spanish professional basketball league; 2) identify the unique profiles of the most successful (i.e., playoff contenders) elite basketball teams. To address aim 1, game-related statistics of all teams were examined over six seasons to identify the evolution of performances. To address aim 2, the playing patterns of teams that most frequently reached the play-off series were studied by non-metric multidimensional scaling model.

## 2. Materials and Methods

### 2.1. Sample and Variables

Archival data were obtained from publicly accessible and official Asociación de Clubes de Baloncesto (ACB or Spanish professional basketball league) records (available at http://www.acb.com/resultados-clasificacion/ver) for 2426 regular season matches played during the period 2009–2010 to 2016–2017. The match-related statistics or team performance indicators analysed included: 2- and 3-point field goals (both made and missed), free-throws (both made and missed), defensive and offensive rebounds, assists, blocks, fouls, steals and turnovers [13–15]. In order to control for match pace, the match-related performance indicators were normalised according to the number of ball possessions [20]. In addition, the data were categorised based upon teams that did or did not reach the playoff series across each and every season analysed (e.g., those teams that reached playoffs each year were Real Madrid, FC Barcelona, Caja Laboral, Gran Canaria, Valencia, and Unicaja).

### 2.2. Statistical Analysis

A multivariate analytical method was used to uncover dynamical trends of the team performance indicators [17]. Multivariate methods were chosen as they enabled the mapping of the entire team match styles rather than individual indicators that determine inferences based upon sets of models [21].

Furthermore, a multivariate method allowed the capture of the temporal trend, simultaneously accounting for all variables in the dataset [17]. While univariate models (e.g., linear regression) can offer powerful insight into individual team performance indicator variability over time, the multivariate technique used allowed for simultaneous analysis and visualisation of the data. For the current dataset, the non-linear multivariate analysis employed was non-metric multidimensional scaling (nMDS). The nMDS plot is a way to condense information from multidimensional data (multiple variables) into a 2D (2 dimensions) representation or ordination. In this ordination, the closer two points were, the more similar the corresponding samples were with respect to the variables that went into making the nMDS plot. The closer the points are together in the ordination space, the more similar their playing style. Unlike other ordination techniques that rely on (primarily Euclidean) distances, such as Principal Coordinates Analysis, nMDS uses rank orders, and thus is an extremely flexible technique that can accommodate a variety of different kinds of data.

The nMDS method has been used extensively across many quantitative science fields such as ecology [22,23], bioinformatics [24], and linguistics [25]. Fundamentally, nMDS analyses the similarity of an n × p data matrix (for each season) where the n rows represent the samples (e.g., teams) and the p columns (e.g., performance indictors) represent the variables measured within each sample. From the n × p data matrix, a distance matrix is calculated based on the ranked similarities. Ranked similarities are preferred when no assumptions are made about the underlying distribution of the data.

Using the full suite of performance indicators, a matrix of (dis)similarity scores was created using the meta MDS function from the "vegan" package [26]. The Bray–Curtis (dis)similarity measure was then used to calculate the (dis)similarity matrix. The (dis)similarity matrix was then plotted in two dimensions and convex hulls highlighted the team match profiles grouped by season. All data were plotted together, with separate team ordinations also plotted to show the temporal change of each team's match profile over the eight-season period. The relationships between the ordination and the individual team performance indicators were visualised by overlaying ordination surfaces. The ordination surfaces were fitted using generalised additive models employing an isotopic smoother via thin-plate regression splines following the procedure of Woods et al. [27]. The season, average match-activity profile (dis-similarity scores) were plotted for the winning and losing playoff teams over the eight seasons. This enabled a comparison between the "dominant" (i.e., Real Madrid and Barcelona) teams' profile within each season relative to the remaining teams within the league. It was possible that the strategies implemented by these dominant teams contributed to a league-wide evolution [16]. One-way, repeated-measures, analysis of variance (ANOVA) with Bonferroni post-hoc tests were used to compare group (season) differences. Statistical significance was set at $p < 0.05$. Effect size (ES) was calculated to determine the pairwise comparisons among seasons using the Cohen's d statistic (d), where an effect size of d < 0.2 was considered small, d = 0.21–0.50 moderate, d = 0.51–0.80 large, and d ≥ 0.80 very large [28]. Effect sizes, and subsequent 90% confidence intervals (90% CI) were calculated in the "MBESS" package [29] with all analyses being undertaken using R version 3.6.2 (R Core Team, 2015).

## 3. Results

### 3.1. Dynamics of Team's Performance

As shown in Figure 1, there was a rapid and sustained increase (statistically significant differences $p < 0.05$ among seasons) in the number of 3-point field goals made (d = 0.61; 90% CI = 0.23; 1.37), 3-point field goals missed (d = 0.72; 90% CI= 0.35; 1.46) and assists (d = 1.27; 90% CI = 0.82; 1.86) from the 2012 season onwards. In contrast, the number of 2-point field goals made (d = 0.21; 90% CI = −1.25; 2.02) and missed (d = 0.27; 90% CI = −0.52; 0.92) declined during the similar period (i.e., 2013 to 2016 seasons, Figure 1). Over the entire sample period, the trend in the number of free throws made (d = 0.36; 90% CI= −0.17; 0.58) and missed (d = 0.43; 90% CI = 0.17; 0.81), offensive rebounds (d = 0.66; 90% CI = 0.35; 1.16), defensive rebounds (d = 0.78; 90% CI = 0.41; 0.91), fouls (d = 0.21;

90% CI = −1.07; 0.56), blocks (d = 0.87; 90% CI = 0.29; 1.56), steals (d = 0.56; 90% CI = 0.19; 0.84) and turnovers (d = 1.25; 90% CI = 0.62; 1.72) remained relatively stable without statistically ($p > 0.05$) significant differences among seasons (Figure 1).

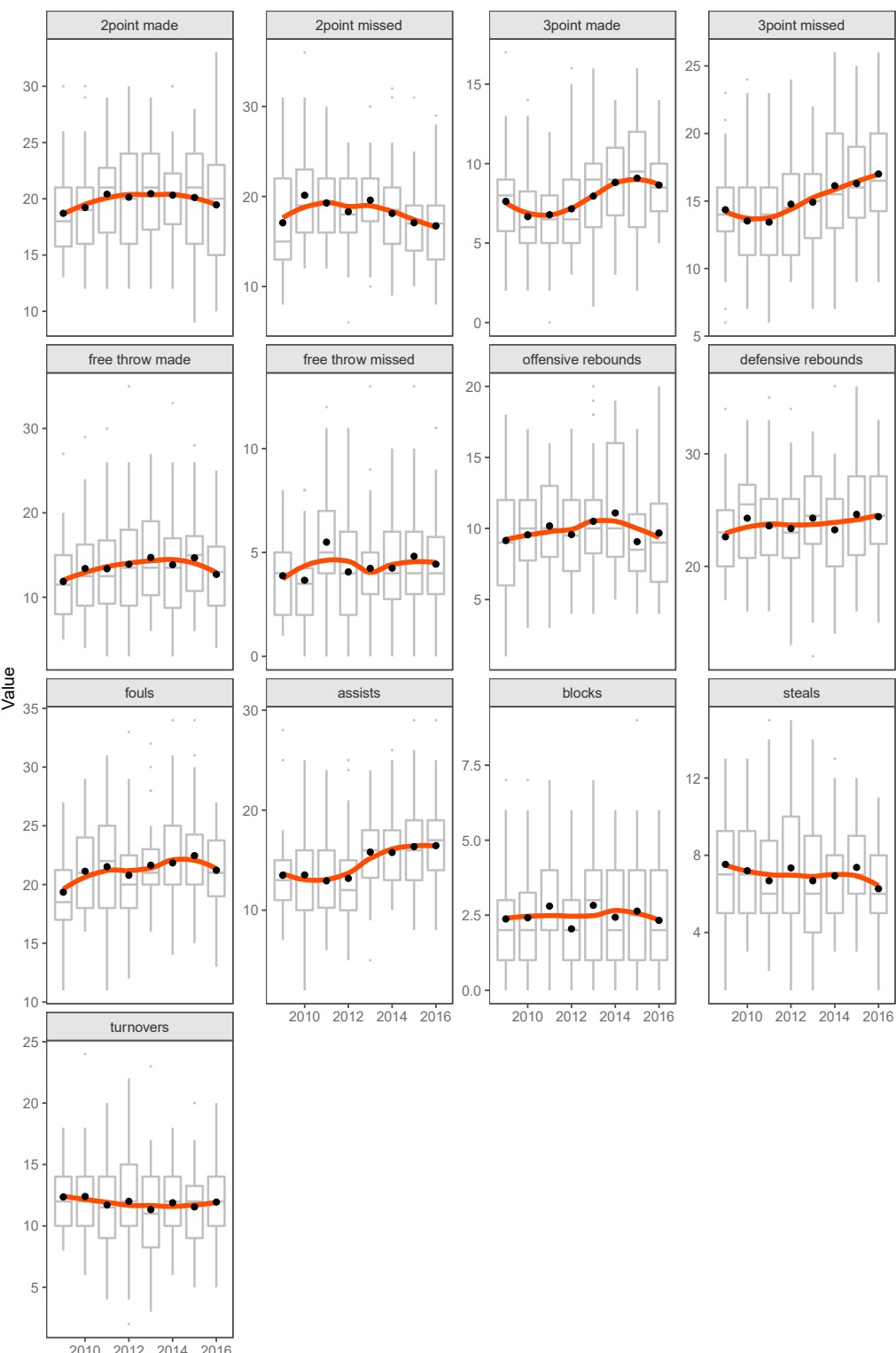

**Figure 1.** Box and whisker plots with median values interquartile ranges and outliers for game-related statistics across the seasons under analysis. Each box is the mean value for each season. The line represents the regression line.

*3.2. Multivariate Team Performance Dynamics*

The (dis)similarity matrix solution was reached after eleven runs (stress = 0.22, rmse (root mean squared error) = $7.9 \times 10^4$, maximum residual = $5.8 \times 10^3$). The ordination plot showed general similarity over the observational period for all teams given their clustered positioning on the plot (Figure 2). Specifically, the dynamics of the competition evolved minimally season-to-season as the overlapping part of the ordination plot and convex hulls indicated that team profiles remained generally similar over the observational period.

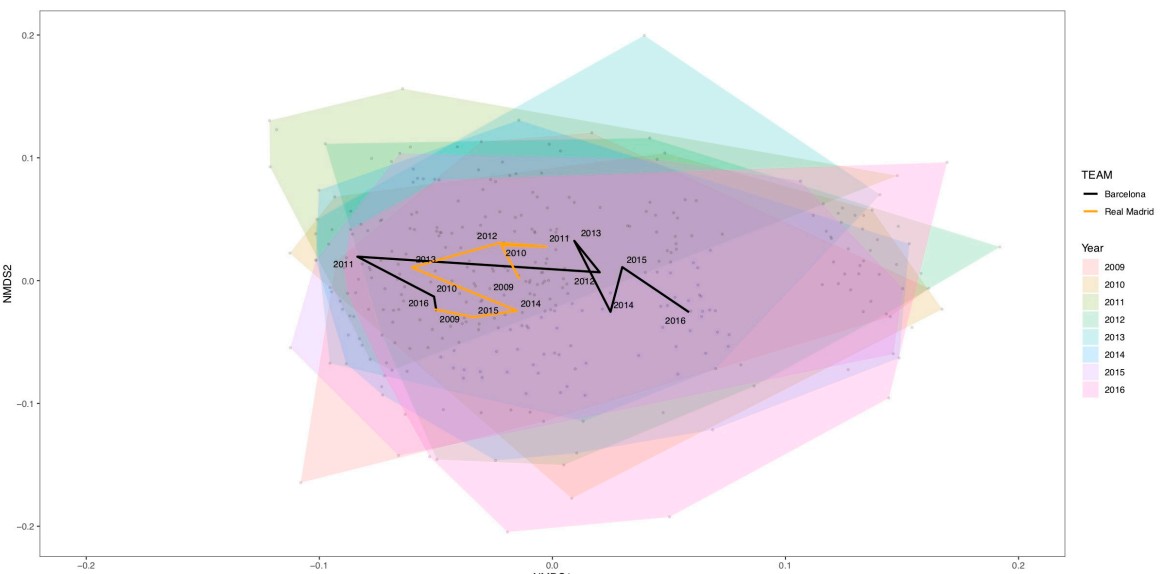

**Figure 2.** The ordination plot using non-metric multidimensional scaling (nMDS) of a distance matrix calculated according to the team game-related statistics of all teams throughout an eight-season period.

The performance profiles of the dominant teams in the ACB (e.g., Real Madrid and Barcelona) generally displayed different playing styles. The ordination plot of each team is presented in Figure 3. These plots indicated that each team displayed a unique profile path throughout the eight seasons. The direction that team profiles were progressing across the ordination surface demonstrated relatively different trends amongst teams. It was worth noting that Caja Laboral presented the most similarity of playing patterns with large season-to-season similarity relative to the remaining teams (Figure 3).

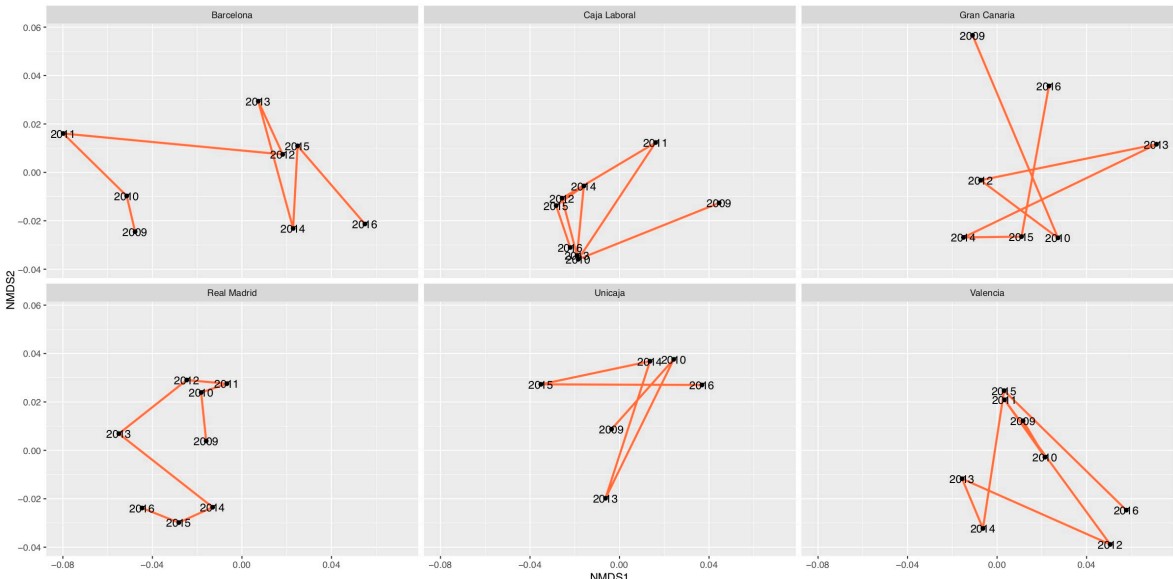

**Figure 3.** nMDS plot for teams that played the playoff series throughout the seasons under analysis.

## 4. Discussion

The aim of the current study was to model the performance of basketball teams' match play considering match-related statistics across eight seasons, and for the most successful or dominant teams during the period under study. As the first examination of the long-term evolution of several professional basketball team's performance [17], the current findings indicated that most match-related performance indicators failed to evolve over time, except for 3-point field goals and assists (2012–2017) which increased while 2-point field goals (2013–2016) decreased. In particular, these indicators reflected the growing impact of shot selection on ball possessions and match outcome [5,12]. Previously, Jaguszewski [15] investigated the evolution of 3-point field goal percentage during the last 15 seasons in the NBA (2004–2005 to 2018–2019) and reported that this match-related statistic had increased its importance over time, but it was not the most discriminative indicator for winning [15]. Additionally, he indicated that the player's ability to create and successfully make 3-point field goals was crucial to disturb defensive actions (i.e., defensive helps) and create open spaces for offensive actions [15]. The current results within the Spanish professional league (ACB) provide further evidence of the evolution of the use and impact of 3-point field goals in professional basketball leagues around the world.

The identified trends for made and missed 3-point field goals also reflected the greater use of this action during matches. These results may demonstrate that match pace (rhythm) had also increased over time and confirmed prior work that defensive actions (e.g., securing defensive rebounds) were important to allow greater fast-break and transition opportunities [6]. Further, the increased number of assists over time supported that teams exhibited greater teamwork [30] and shooting success possibly via short attacking actions [10,12]. Previously, winning teams were reported to be more disciplined when performing collective tactical actions (i.e., teamwork) during both short and long attacks (i.e., increasing scoring opportunities) [31]. The current results can provide further basis for these group-tactical behaviours that lead to open 3-point shots during set plays and transitions (e.g., match pace variations and inside–outside passes), and easy scoring during fast breaks (e.g., quick actions after defensive rebounds) [32–34].

While many match-related indicators evolved over time, defensive rebounds were steady and had a constant impact on match outcome across seasons. This result was unexpected given that this performance indicator was considered to be one of the main contributors to winning basketball matches across a range of competitions [12–14,31]. In particular, this indicator was reported to be closely related

to shooting effectiveness with defensive rebounding a key action to secure the ball from an opponent's mistake and then quickly play to counterattack and score [5]. Therefore, defensive rebounds may have remained stable during the period 2010–2017 due to the reciprocal trends in both 2- (i.e., decrease) and 3-point (i.e., increase) field goals. Future changes in shooting success within the league may alter defensive rebounds with coaches encouraged to monitor continually this indicator for strategic team planning including player recruitment.

Interestingly, in the current study steals and turnovers remained stable over time, which was in contrast with prior work of winning teams in international basketball tournaments [13,14]. Potentially, competition type (national vs. international) may explain these contrasting results with different styles of match play exhibited by teams when winning and performing at the elite level [33]. Future work examining national and/or international competitions would clarify the impact of competition level on the evolution of the competition.

A key focus of the current study was to examine the evolution of teams within the Spanish League, especially those six teams that contested all playoff series. Each team presented a unique performance profile with different trends across the eight seasons. The dominant teams, Real Madrid and FC Barcelona exhibited unique match play patterns across seasons with Real Madrid's pattern of most interest as it varied very little (steady season-to-season), remaining to the left of the nMDS origin. Subsequently, Real Madrid's playing pattern may be considered as a benchmark given their ongoing success and minimal evolutionary change—what they are doing is successful (e.g., variation of performances across the seasons with more ball possessions and 3-point field goals as the seasons progressed). Comparatively, FC Barcelona exhibited a more variable pattern that was located on both sides of the nMDS origin with adaptations to each season resulting in a larger change and potentially less long-term success. Additionally, Caja Laboral generally clustered in the middle of the ordination surface and followed a unique pathway that was relatively different to other teams. In fact, the Caja Laboral team used the same player line-up and stable tactical system promoted by the same coach with performance trends (e.g., less ball possessions and defensive actions such as fouls committed and defensive rebounds) similar during the 2012, 2014 and 2015 seasons, and again during the 2010, 2013 and 2016 seasons. Despite these team differences, the dominant patterns were reflective of the most successful ACB teams over the long term with each demonstrating assertive offensive (i.e., high-pressure attacks) and defensive (i.e., balls recovered and securing rebounds) play that increased match pace [34]. Furthermore, these dominant patterns indicated the successful adaptations of match play according to the opponent and the type of match (i.e., fast- or slow-paced) during single and multiple seasons [34]. Specifically: (i) most successful teams (i.e., final winners of the league, Real Madrid and FC Barcelona) maintained their match play patterns across seasons with minor modifications and better adapted to the requirements of the league [17]; (ii) teams that classified for playoff series performed differently than the remaining teams and potentially less predictable and more adaptive to competition evolution [16]; (iii) the unique patterns indicated each team's idiosyncrasy during a national competition and reflected their strengths and weaknesses across the seasons [17]. Using the displayed evolutionary trends, coaching staff and performance analysts could focus their own training and recruitment on the most important performance indicators to succeed during competition (e.g., greater 3-point field goals made and assists), while also scouting opponents' long-term match playing patterns for strategic actions [16,17]. Subsequent changes could lead to team behaviours that favour long-term success [16].

The current study has provided evidence of the minimal evolution for an elite, professional basketball league with some limitations that need to be acknowledged. First, the current analysis was limited to eight seasons with analysis of more seasons (e.g., the last decade) potentially identifying long-term evolutionary specificities of the Spanish basketball league (ACB) and more successful or dominant teams. Second, standard match-related indicators were examined like prior work [13,14] with the production of new secondary match and ball possession indicators possibly providing greater ability to examine success and the evolution of basketball teams. Thirdly, contextual factors

(e.g., game location) were not considered in the current study with further research needed to identify the long-term performance trends of winning and losing teams in accordance with contextual factors. Lastly, nMDS was the primary tool to demonstrate the evolution of teams with a combination of statistical and analytical methods potentially providing greater modelling and classification/prediction of performances in elite basketball.

## 5. Conclusions

This study confirmed that most performance indicators (except 2-point and 3-point field goals, and assists) were stable across eight seasons within the Spanish professional basketball league. Minimal season-to-season evolution had occurred for most teams with the predominant team (Real Madrid) displaying a unique and established winning pattern compared to other teams. The current study identified the evolutionary performance trends of Spanish professional basketball teams with further modelling of these performances likely providing future competition and training support to coaches.

**Author Contributions:** Conceptualization, M.-Á.G. and A.V.; methodology, M.-Á.G., R.M., and A.V.; software, S.Z.; validation, M.-Á.G., R.M., A.V. and S.Z.; investigation, M.-Á.G., A.V., and A.S.L.; writing—original draft preparation, M.-Á.G., A.V., S.Z., and A.S.L.; writing—review and editing, M.-Á.G., A.V., S.Z., and A.S.L.; visualization, S.Z.; supervision, M.-Á.G., A.V., S.Z., and A.S.L.; project administration, M.-Á.G. and A.V. All authors have read and agreed to the published version of the manuscript.

**Funding:** This research received no external funding.

**Acknowledgments:** This study was supported by a mobility Grant "Salvador de Madariaga" (PRX18/00098; Ministry of Education, Culture and Sport of Spain) for the corresponding author during a research exchange at James Cook University (Townsville, Australia). The research was also supported by "Programa de ayuda al grupo de investigación psicosocial en el deporte UPM-2019".

**Conflicts of Interest:** The authors declare no conflict of interest.

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
