# Peer review of "The Performance Evolution of Match Play Styles in the Spanish Professional Basketball League"

_applsci, doi:10.3390/app10207056_

Round 1
Reviewer 1 Report
The paper analyzes the dis(similar) evolution performance of all teams, and the dominant basketball team during eight seasons. Match-related statistics were gathered from all regular season matches. Authors claim that the most dominant teams exhibited unique match patterns with the most successful team's pattern.
The authors study the (dis)similarity of teams and reach some interesting conclusions.
Some conclusions are not supported in the paper. For instance: "3-point field-goals made and missed also reflected the greater number of ball possessions during matches". It is not clear how some statements appear along with the document.
The structure of the paper should be included at the end of section 1.
Before section 3. The interval of d (d=0.21-0.50) is not clear. Please use d in ]0.20,0.50] or 0.21 < d <= 0.50. The same for CI in section 3.1.
Rewrite the introduction paragraph of Section 3.
Figure 2: It is not clear when the lines begin. I.e., the point associated with the year 2009. Figure 2 should be better explained. What are NMDS1 and NMDS2? How are they evaluated?
The sentence before Figure 2 does not make sense.
Figure 3: What mean X1 and X2? More details should be added.
In order to improve the paper, a relationship, each year, between the patterns and wins should be considered.
Remarks:
- The references should not use curve-brackets (they are not equations).
- Highlight subsections of section 2.
- Before section 3. Insert a space before 'MBESS.'
-line 184: a bracket is missing.
-Remove one mark before Section 5.
Author Response
Dear reviewer thanks for your detailed review and suggestion to improve the manuscript. We have addressed all the changes/ modification suggested in red letters.
REV.- The paper analyzes the dis(similar) evolution performance of all teams, and the dominant basketball team during eight seasons. Match-related statistics were gathered from all regular season matches. Authors claim that the most dominant teams exhibited unique match patterns with the most successful team's pattern.
AUT.- No response needed.
REV.- The authors study the (dis)similarity of teams and reach some interesting conclusions.
Some conclusions are not supported in the paper. For instance: "3-point field-goals made and missed also reflected the greater number of ball possessions during matches". It is not clear how some statements appear along with the document.
AUT.-Thanks for this comment, we have revised and modified the conclusions accordingly.
REV.- The structure of the paper should be included at the end of section 1.
AUT.- Thanks for your suggestion, we have added a sentence clarifying the structure of the manuscript.
REV.-Before section 3. The interval of d (d=0.21-0.50) is not clear. Please use d in ]0.20,0.50] or 0.21 < d <= 0.50. The same for CI in section 3.1.
AUT.- Done
REV.- Rewrite the introduction paragraph of Section 3.
AUT.- Thanks for your comment, the paragraph was removed to avoid any misunderstanding.
REV.- Figure 2: It is not clear when the lines begin. I.e., the point associated with the year 2009.
AUT.- Done
AUT.- Non-metric Multi-dimensional Scaling (NMDS) is a way to condense information from multidimensional data (multiple variables), into a 2D representation or ordination. In this ordination, the closer two points are, the more similar the corresponding samples are with respect to the variables that went into making the NMDS plot. The closer the points are together in the ordination space, the more similar their playing style. Unlike other ordination techniques that rely on (primarily Euclidean) distances, such as Principal Coordinates Analysis, NMDS uses rank orders, and thus is an extremely flexible technique that can accommodate a variety of different kinds of data.
REV.- Figure 2 should be better explained. What are NMDS1 and NMDS2? How are they evaluated?
AUT.- Thanks for your comment, see the response of the previous comment that clarified the plot.
REV.- The sentence before Figure 2 does not make sense.
AUT.- Thanks for your comment, the sentence was removed.
REV.- Figure 3: What mean X1 and X2? More details should be added.
AUT.- Thanks for your suggestion. We have changed the plot. X1 means NMDS1 and x2 means NMDS2.
REV.- In order to improve the paper, a relationship, each year, between the patterns and wins should be considered.
AUT.- Thanks for your suggestion. However, the analysis of winning and losing teams is a different issue than that covered in the current study (playing patterns evolution in Spanish Basketball). We acknowledge this issue in further research to identify performance trends to win along the seasons.
REV.- The references should not use curve-brackets (they are not equations).
AUT.- Done
REV.- Highlight subsections of section 2.
AUT.-Done
REV.- Before section 3. Insert a space before 'MBESS.'
AUT.-Done
REV.- line 184: a bracket is missing.
AUT.-Done
REV.- Remove one mark before Section 5.
AUT.-Done
Reviewer 2 Report
The article has merit. However, minor revisions must be done before acceptance.
In this paper is necessary to improve the writing of:
- Mathematical notations of confidence interval and effect size (e.g. [number1 ; number2]);
- Notation of power of base ten;
- Section of Statistical analysis, clarifying for that if use the statistical methods. For example, what is the statistical method used to compare the performance between seasons? The cohen’d is the effect size associated to t-student (independent sample or paired sample).
In mathematics the expression "Matrix solution", namely Theory of Matrices, is different from the context applied in this paper.
In section 5. conclusions, it is necessary to improve the writing of main contribution of this study.
Author Response
Dear reviewer thanks for your detailed review and suggestion to improve the manuscript. We have addressed all the changes/ modification suggested in red letters.
REV.- The article has merit. However, minor revisions must be done before acceptance.
In this paper is necessary to improve the writing of: Mathematical notations of confidence interval and effect size (e.g. [number1 ; number2]);
AUT.- Done
REV.- Notation of power of base ten;
AUT.- Done
REV.- Section of Statistical analysis, clarifying for that if use the statistical methods. For example, what is the statistical method used to compare the performance between seasons? The Cohen’d is the effect size associated to t-student (independent sample or paired sample).
AUT.- Thanks for your suggestion. We have mentioned that the ordination surfaces were fitted using generalized additive models employing an isotopic smoother via thin-plate regression splines. We complete the specific step according to the following paper: Carl T. Woods, Sam Robertson & Neil French Collier (2017) Evolution of game-play in the Australian Football League from 2001 to 2015, Journal of Sports Sciences, 35:19, 1879-1887, DOI: 10.1080/02640414.2016.1240879
In addition, we have clarified the statistical analyses undertaken (repeated measures ANOVA and Bonferroni post hoc test).
REV.- In mathematics the expression "Matrix solution", namely Theory of Matrices, is different from the context applied in this paper.
AUT.-Thanks for your comment. Yes, we understand that both concepts do not mean the same. In particular, our method accounts for the “data matrix” used for each season. We added this information within the methods section to clarify this issue (line 102).
REV.- In section 5. conclusions, it is necessary to improve the writing of main contribution of this study.
AUT.-Done
Reviewer 3 Report
The paper is written very well, and the analysis is performed in a good methodological manner. Here are some comments to improve presentation.
- There are many repetitions when presenting the results. The summary in line 221 is re-expressed in many places in the article in quite a redundant and profuse way which makes it difficult to follow.
- The use of "(dis)similairty" is very confusing. Say clearly which patters are similar and whic are not.
- Figure 2 - you say in line 138 that it contains plots for all teams, but it only contains for two. Why these two? what's special about them? what does the drawing tell you?
- Lines 149-150 - something went wrong there
- Line 164-165: say what these patterns are.
Some references of works that analyze patterns in football teams using a different way, not using multivariate analysis. Can be interesting to try them also and perhaps cite them in your paper.
- Neuman Y., Vilenchik D., 2019. Modeling Small Systems through the Relative Entropy Lattice. IEEE Access.
- Neuman Y., Israeli, N. Vilenchik, D. and Cohen, Y. 2018. The Adaptive Behavior of a Soccer Team: An Entropy-Based Analysis. Entropy.
Author Response
Dear reviewer thanks for your detailed review and suggestion to improve the manuscript. We have addressed all the changes/ modification suggested in red letters.
REV.- The paper is written very well, and the analysis is performed in a good methodological manner. Here are some comments to improve presentation.
AUT.-Thanks for your revision and useful comments
REV.- There are many repetitions when presenting the results. The summary in line 221 is re-expressed in many places in the article in quite a redundant and profuse way which makes it difficult to follow.
AUT.-Thanks, the text was revised accordingly.
REV.- The use of "(dis)similairty" is very confusing. Say clearly which patters are similar and which are not.
AUT.-Thanks for your suggestion, the terms were revised accordingly.
REV.- Figure 2 - you say in line 138 that it contains plots for all teams, but it only contains for two. Why these two? what's special about them? what does the drawing tell you?
AUT.-Thanks for your comment, we included only those 2 dominating teams that reached the finals over the 8 seasons. We wanted to show their playing patterns according to the season with different trends for both teams along the seasons and between them.
REV.- Lines 149-150 - something went wrong there
AUT.-Thanks, the sentence was deleted.
REV.- Line 164-165: say what these patterns are.
AUT.-The sentence was clarified.
REV.- Some references of works that analyze patterns in football teams using a different way, not using multivariate analysis. Can be interesting to try them also and perhaps cite them in your paper.
- Neuman Y., Vilenchik D., 2019. Modeling Small Systems through the Relative Entropy Lattice. IEEE Access.
- Neuman Y., Israeli, N. Vilenchik, D. and Cohen, Y. 2018. The Adaptive Behavior of a Soccer Team: An Entropy-Based Analysis. Entropy.
AUT.-Thanks for your suggestion, both references were included.
Round 2
Reviewer 1 Report
The authors have addressed most part of my concerns. Minor remarks: - line 16: consider changing "was" to "is" - line 46, consider inserting a comma after "winning" - lines 99-106: NMDS or nMDS? - line 131: What you mean by "d=0.51-0.80"? - line 132: [29] not (29] - line 220: "(e.g.)"? Does not make sense - line 271: remove the last markAuthor Response
Dear reviewer thanks for your review and suggestion of minor issues that allow us to improve the manuscript. We have addressed all the changes/ modification suggested in red letters.
REV.- Minor remarks: - line 16: consider changing "was" to "is"
AUT.- Done.
REV.- line 46, consider inserting a comma after "winning"
AUT.- Done.
REV.- lines 99-106: NMDS or nMDS?
AUT.- nMDS, the change was made accordingly.
REV.- line 131: What you mean by "d=0.51-0.80"?
AUT.- These values mean “large”, we added the word.
REV.- line 132: [29] not (29]
AUT.- Done.
REV.- line 220: "(e.g.)"? Does not make sense
AUT.- Thanks, we deleted the bracket.
REV.- line 271: remove the last mark
AUT.- Done.